# Machine Learning-Based Coastal Terrain Classification in Tropical Regions Using Multispectral UAV Imaging: A Comparative Study of Random Forest and SVM Models

Anonymous Full Paper
Submission 34

## Abstract

Advances in various technologies and machine learning (ML) are transforming the field of remote sensing. This study proposes an ML-centered methodology for classifying coastal terrain in tropical coastal regions using multispectral unmanned aerial vehicle (UAV) image inputs. The objective is to identify suitable ML algorithms for analyzing multispectral images on limited hardware. Multispectral images of the study area were collected using a DJI Mavic 3M UAV in March 2023. K-means clustering was implemented to assist in coastal terrain identification, and the labeled data were used to train pixel-based Support Vector Machine (SVM) and Random Forest (RF) models utilizing a 5-fold block cross-validation scheme. The results showed that the optimized RF model outperformed the SVM model across most metrics. Despite this, the SVM model showed potential for live image classification due to its smaller size and quick classification speed. Additionally, the optimized models effectively classified images from areas set as an independent hold-out test set, demonstrating the applicability of ML in this type of remote sensing problem.

## 1 Introduction

Climate change has significantly impacted coastal ecosystems, leading to their degradation through rising temperatures, ocean acidification, and urban encroachment [1]. Given the importance of these ecosystems for biodiversity and biomass production, urgent measures are needed to mitigate the effects of anthropogenic climate change.

Traditional environmental assessment methods rely on on-site teams to collect data on species populations, soil and water quality, and human settlements, but these methods are labor-intensive and time-consuming. Modern approaches use remote sensing technologies like satellite imagery, multispectral sensors, and LiDAR, allowing for faster and more accurate environmental monitoring. Unmanned aerial vehicles (UAVs) have further enhanced data collection by providing high-resolution images that bridge the gap between satellite data and on-site surveys. Processing this data involves advanced computational techniques, including machine learning algorithms, which facilitate rapid and detailed analysis of environmental conditions.

Live image segmentation from UAVs is an exciting and emerging area of research in machine learning. However, several challenges must be addressed to make machine learning viable for live or near-live classification. First, models need to be compact enough to run on limited onboard processing power. They must also offer low-latency performance, as faster classification times are preferable, and be power-efficient to extend flight duration. Additionally, the model may need to share onboard resources with image preprocessing tasks, such as correcting for image warping or other artifacts [2, 3]. Suitable hardware options for this task include devices like the Arduino Portenta H7, ESP32-CAM, and Raspberry Pi Zero 2 W, which offer memory capacities of 16 MB, 4 MB, and support for an SD card, along with RAM sizes of 8 MB, 4 MB, and 512 MB, respectively [4–6].

In the context of traditional ML approaches to object-based and pixel-based classification, Support Vector Machine (SVMs) and Random Forests (RFs) are among the most popular for use in remote sensing as powerful machine learning algorithms with distinct strengths and weaknesses [7–9]. SVMs excel in high-dimensional spaces where the number of features exceeds the number of samples and robust to overfitting, especially in cases where the data is sparse [10]. However, they can be computationally intensive, particularly with large problem sizes, and their performance relies heavily on the careful tuning of hyperparameters, which often involves tedious and time-consuming experimentation and iterative adjustments [11].

On the other hand, RFs are versatile and easy to implement, providing good performance across a wide range of datasets without much need for tuning [12]. They handle large datasets efficiently and are capable of capturing complex interactions between features. However, RFs can sometimes struggle with overfitting, particularly if the number of trees is not sufficiently large, and they can be less effective than SVMs in very high-dimensional spaces. Additionally, RFs tend to require more computational resources as the number of trees grows.

## 2 Methodology

### 2.1 Data Collection

The study was conducted in an 8-hectare area, West of the municipality of Lian, Batangas province of the Philippines. The inland region consists of uneven ground covered by various mangroves, bushes, and grasses. This transitions into a shallow sandbar that extends about 150 meters westward into the sea. Within this area, there are patches of aquatic vegetation and mangroves before the landscape changes to a deeper and rockier region. The dataset analyzed in the present study was obtained through a single aerial survey campaign in March 2023 that began at noon. The weather on the day was fair with little cloud coverage.

The UAV used for data acquisition was the DJI Mavic 3M, manufactured by SZ DJI Technology Co., Ltd., based in Shenzhen, China. It is equipped with a high-resolution 4K RGB alongside a multispectral camera. The imaging capability of the UAV encompasses a wide spectrum of wavelengths, including Green (560 ± 16 nm), Red (650 ± 16 nm), Red Edge (730 ± 16 nm), and Near-Infrared (860 ± 26 nm), enabling the detailed capture of vegetative and geographical features with high spectral resolution [13]. Each pixel within the image corresponds to a spatial resolution of 2 cm, thereby facilitating the extraction of detailed information at a fine scale.

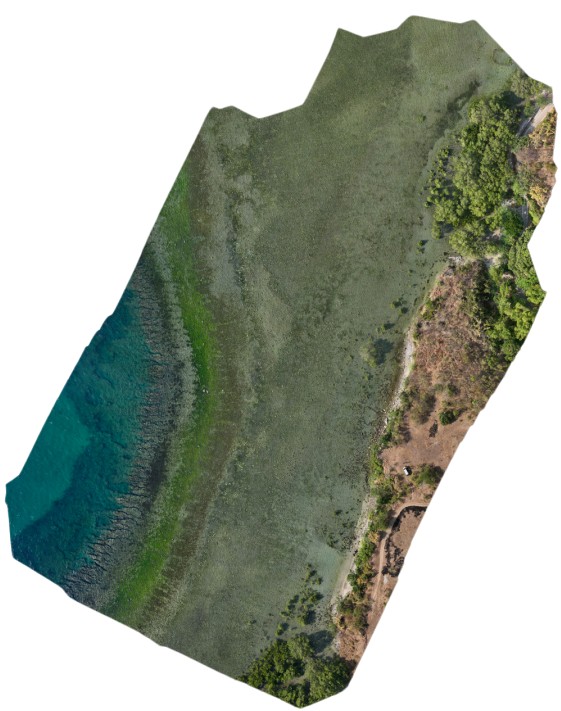

**Figure 1.** Orthomosaic of the region of interest.

### 2.2 Data Processing

The images that were captured were combined to create orthomosaics through the use of onboard software. These orthomosaics encompass various maps such as RGB, Normalized Difference Vegetation Index (NDVI), Green Normalized Difference Index (GNDVI), Normalized Difference Red Edge (NDRE), and Leaf Chlorophyll Index (LCI). The constructed orthomosaic was just under 50,000,000 pixels large. Subsequent data operations were carried out using the multispectral vegetation index (VI) and the multispectral images instead of the RGB images. The unsupervised and supervised algorithms were both implemented using Python.

All machine learning model training and testing was conducted using the free tier of Google Colab, which featured 12.7 GB of RAM [14]. This resource limitation played a significant role in determining the final optimized model. In addition to traditional metrics such as accuracy, precision, recall, and F1-score, training times also factored into the decision-making process: in cases where two models demonstrated comparable performance, the model with the shorter training time was chosen.

### 2.3 Definition of Training Labels

Features were identified by implementing k-means clustering on each of the VIs from $k = 2$ to $k = 8$. A mini-batch algorithm was chosen to reduce the computation time. Each combination of a VI and the $k$ number of clusters was assessed to determine possible terrain types. This assessment was based on both the cluster's silhouette score and a qualitative comparison to the cluster's corresponding region in the RGB image. These were then associated with a terrain type in the image such as "terrestrial vegetation" or "sublittoral zone". The training labels on pixels were then manually adjusted and reassigned to resolve overlaps between clusters or to align them with the correct terrain type based on domain experts.

The silhouette score is a metric used to measure the quality of clusters in a clustering algorithm. It provides an indication of how well each data point lies within its cluster relative to other clusters [15]. The resulting value ranges from -1 to 1, where a value close to 1 indicates that the point is well clustered, with the data point being much closer to points in its own cluster than to points in other clusters. A value close to 0 indicates that the point lies on the boundary between clusters, while negative values suggest that the point may have been assigned to the wrong cluster [16]. By averaging the silhouette coefficients of all points in a dataset, one can obtain an overall measure of cluster quality, where higher average silhouette scores simply better-defined and more

175 distinct clusters. It is mathematically expressed as

$$s_i = \frac{b_i - a_i}{\max(a_i, b_i)} \quad (1)$$

177 where for a data point $i$, $a_i$ denotes the distance
178 between a data point and its assigned centroid while
179 $b_i$ denotes the distance to the closest centroid be-
180 longing to a different cluster.

## 2.4 Coastal Terrain Identification

Through the unique combinations of $k$ and the VIs,
6 terrain classes were identified as seen in Figure
3. NDRE clustered with $k = 2$ was used to identify
class 0, the "sublittoral zone". This corresponds to
the deeper submerged areas of the image. In these
regions, further features are difficult to isolate due

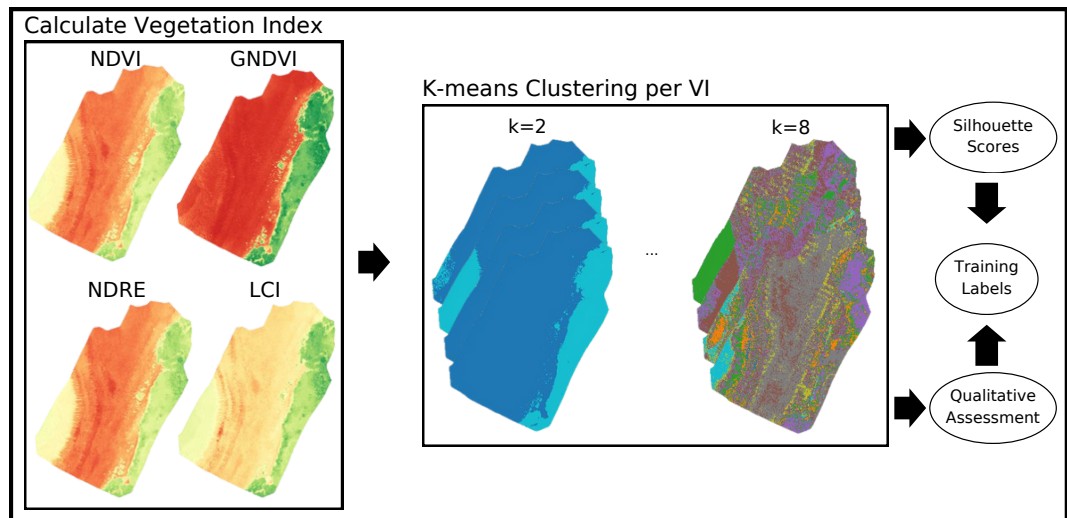

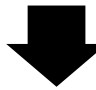

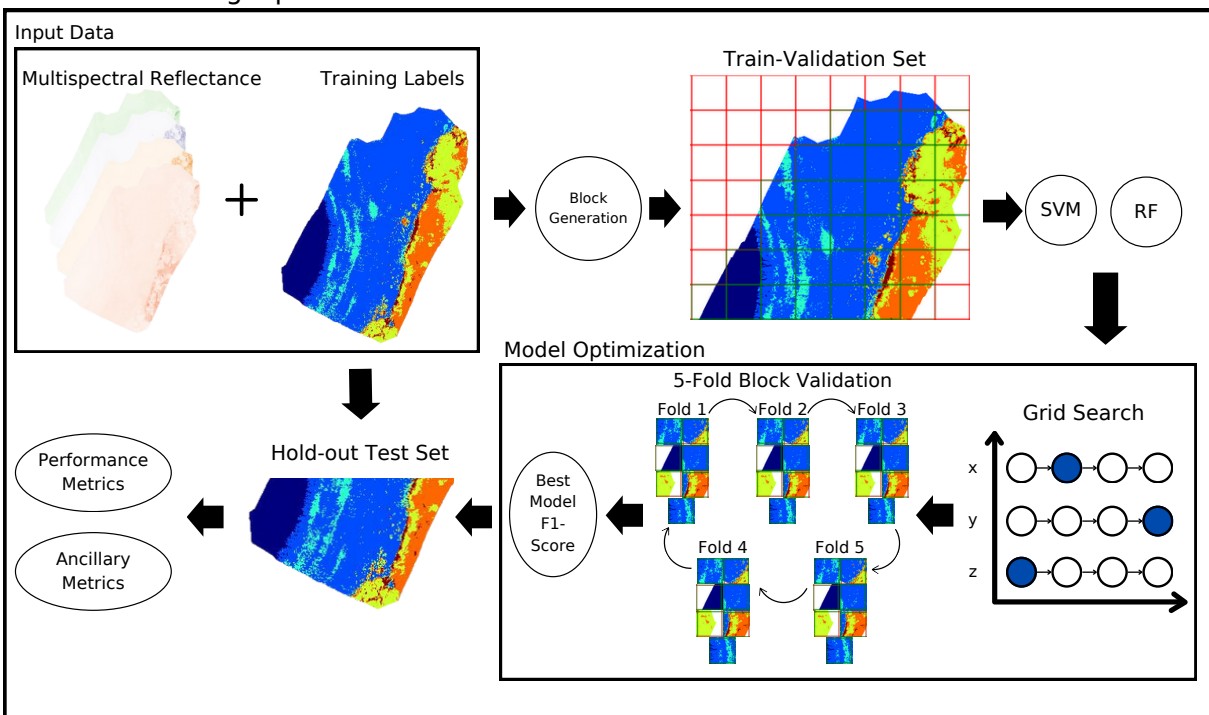

**Figure 2.** Graphical representation of feature extraction through implementing k-means clustering on VIs (top) and machine learning pipeline using hold-out test set and a block k-folds cross-validation scheme with grid search hyperparameter optimization (bottom).

to the depth of the water.

LCI clustered with $k = 5$ was then used to identify classes 1 and 2, "shallow water" and "shallow bare zone" respectively. The "shallow water" cluster represents areas with shallow water and some algal content, while the "shallow bare zone" refers to submerged areas without significant photosynthetic activity.

NDVI clustered with $k = 4$ was used to detect class 3, the "terrestrial vegetation". It consists of trees, bushes, and grasses.

GNDVI clustered with $k = 2$ was used to isolate class 4, labeled "bare land". This refers to regions on the land with little to no vegetation.

Lastly, GNDVI clustered with $k = 5$ was used to isolate class 5, "shadows and rocks" cluster. This is a region where labeling is challenging due to shadows cast by tree canopies or the presence of rocks.

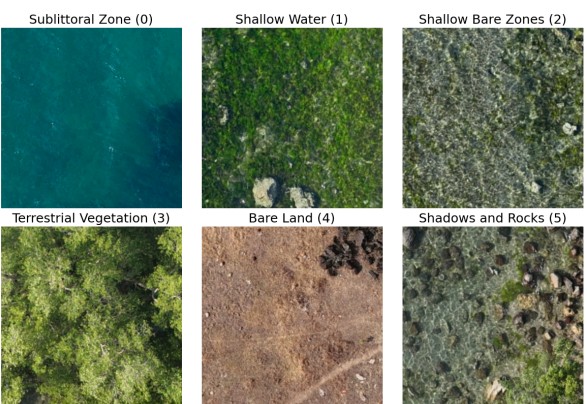

**Figure 3.** Identified classes from k-means clustering. These six classes encompass the terrain types found.

## 2.5 Implementation

SVM and RF models were trained on the layered multispectral bands. These were chosen as the model inputs as VIs require processing and context of the larger image to effectively normalize values whereas model's trained directly on the multispectral bands will be able to classify immediate instances taken by the multispectral camera. Hyperparameters were optimized using a Grid search approach. Grid search finds the optimized hyperparameters of an algorithm using a specified list of values for each hyperparameter. A model is then trained for every possible combination of hyperparameters with the optimized model resulting from the combination that yielded the highest F1-Score. This metric was chosen as the primary metric as it accounts for the misclassification of minority classes that may be underrepresented due to the proportion of labels in the image.

A 5-fold block cross-validation with a separate hold-out test set was used to validate the model. The

image was first separated into a training-validation set in the North with the rest being separated as a hold-out test set as can be seen in Figure 2. The training-validation set was then divided into 34 image blocks equivalent to a 20x20 meter area each. These blocks are then distributed between an $n$ number of subsets or folds. The model is then trained on the $n - 1$ folds of data with the remaining fold being used as a validation set. The process is then repeated, cycling through the various possible validation folds. These results are then averaged to provide an understanding of the performance of the particular model [17]. In the particular case of a 5-fold cross-validation scheme, 80% of blocks at any given time are used as the training data while 20% remains for validation. This is then cycled such that all subsets of 20% are used for validation of the model's performance.

# 3 Results

## 3.1 Machine-Labeled Maps

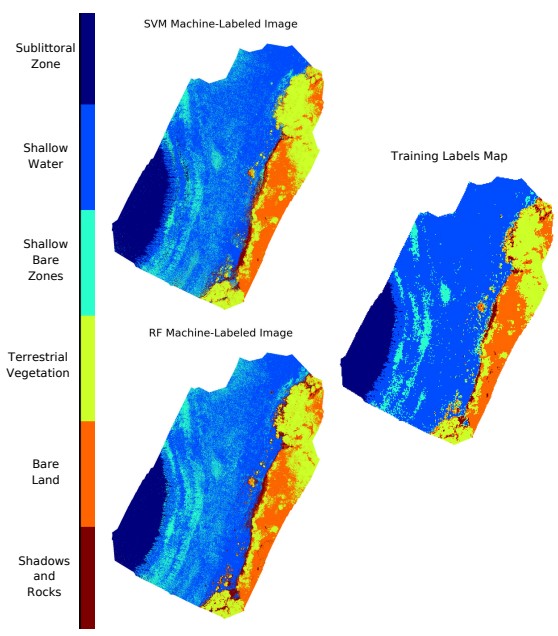

**Figure 4.** Comparison of machine-labeled maps. Shallow bare zones are more prevalent in the machine-labeled maps as compared to the training labels.

Displayed in Figure 4 are the terrain type maps generated by the trained SVM and RF models. Upon initial visual inspection, it was seen that both models were able to label the image similar to the training image. As seen in the figure, shallow water represents the majority of the image with sublittoral zone, shallow bare zones, terrestrial vegetation, and bare land being smaller classes of similar size. As expected, the shadows and rocks is seen as the smallest

minority class of the image.

However, some immediate differences are obvious between the machine-labeled maps and the training labels. Both machine learning models appear to assign pixels to the shallow bare zone terrain type at a rate higher than the training labels. These manifest as more spread out throughout the image as opposed to the tighter concentrations found in the training labels. A second observation is the spread of the shadows and rocks clusters in the SVM-map being much more prevalent along the coastline as opposed to the training map and RF-labeled map.

## 4  Discussion

### 4.1  Experiments with Forests

For the purposes of minimizing file size and training time in RF models, particularly close attention was given to the number of trees and the maximum depth of trees in the models. Training time was seen to increase linearly with both the number of trees and the maximum depth of the trees. Between these two, the maximum depth of trees was the more important factor in determining model performance.

The final hyperparameters chosen for the random forest model reflect a sparse forest of only 20 trees with a depth of 30. Forests with a greater number of trees only a minor amount of improvement in the validation set while extending training by many more minutes.

### 4.2  Optimized Models

**Table 1.** SVM classification report on the independent test set. Generally good performance across terrestrial terrain types and the sublittoral zone.

| SVM Accuracy: 0.85 | Precision | Recall | F1-Score |
| --- | --- | --- | --- |
| Sublittoral Zone | 0.91 | 0.98 | 0.94 |
| Shallow Water | 0.90 | 0.82 | 0.86 |
| Shallow Bare Zones | 0.78 | 0.77 | 0.78 |
| Terrestrial Vegetation | 0.95 | 0.97 | 0.96 |
| Bare Land | 0.88 | 0.92 | 0.90 |
| Shadows and Rocks | 0.20 | 0.33 | 0.25 |
| Macro Average | 0.77 | 0.80 | 0.78 |
| Weighted Average | 0.86 | 0.85 | 0.85 |

The SVM model performed relatively well with an accuracy of 0.85. When taking their weighted average (that is the average of each metric weighted by its number of samples) the Precision, Recall, and F1 scores across all classes are 0.86, 0.85, and 0.85 respectively. These scores drop when considering the macro average which considers the scores of each class as being of equal weight. Using this method of averaging, the scores drop to 0.77, 0.80, and 0.78

suggesting that there is a higher incidence of false negatives with the model.

Looking into the individual metrics per class, we see that the model's performance in identifying the sublittoral zone, shallow water, terrestrial vegetation, and bare land classes is good. However, there is a high rate of false negatives in the shallow bare zone and shadows and rocks regions with their Recall scores being only 0.77 and 0.33 respectively.

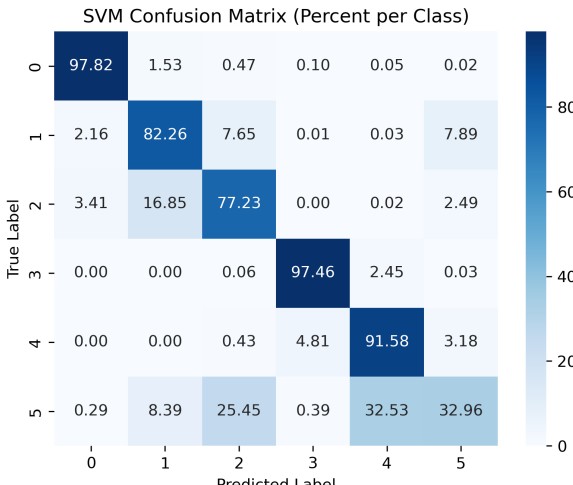

**Figure 5.** SVM confusion matrix on the independent test set. Minor misclassification observed between the "shallow water" and "shallow bare zones" with heavy misclassification in "shadows and rocks".

Referring to the SVM model's confusion matrix in Figure 5., it is seen that the sublittoral zone, terrestrial vegetation, and bare land terrain types are accurately classified. However, the shallow water and shallow bare zones offer a challenge being commonly mistaken for each other resulting in correct predictions only 82.26% and 77.23% of the time and misclassification of shallow water for shallow bare zones at 7.65% with the reverse occurring more often at 16.85%. The most prevalent case of erroneous classification manifests in the shadows and rocks cluster with only 32.96% of the true labels being correctly predicted thereby underscoring the challenges in classifying this terrain type. This can be explained by this terrain type's presence in both aquatic and terrestrial portion of the image as reflected by the 25.45% misclassification into the shallow bare zones and 32.53% in the bare land.

The RF model exhibited superior performance compared to the SVM model, achieving an accuracy of 0.98. Furthermore, it demonstrated strong performance across all metrics in both macro and weighted averages. Upon analyzing its performance within each class, it maintained high accuracy for shallow water, shallow bare zones, terrestrial vegetation, bare land, and shadows and rocks. The only exception was the shallow bare zones and shadows

and rocks, which exhibited a small amount of misclassification, indicated by a Precision score of 0.94. Nonetheless, the overall performance of the model remained commendable.

**Table 2.** RF classification report on the independent test set. Minor errors in "shallow water" and "shadows and rocks".

| RF Accuracy: 0.98 | Precision | Recall | F1-Score |
|---|---|---|---|
| Sublittoral Zone | 0.99 | 0.98 | 0.99 |
| Shallow Water | 0.99 | 0.97 | 0.98 |
| Shallow Bare Zones | 0.94 | 0.99 | 0.96 |
| Terrestrial Vegetation | 0.99 | 0.99 | 0.99 |
| Bare Land | 0.98 | 0.99 | 0.98 |
| Shadows and Rocks | 0.89 | 0.99 | 0.93 |
| Macro Average | 0.96 | 0.98 | 0.97 |
| Weighted Average | 0.98 | 0.98 | 0.98 |

Referring once again to the model's corresponding confusion, it is observed that in nearly all of the classes, the majority of pixels lies along the diagonal with no misclassification exceeding 2.5% of pixels. The RF model largely prevents the frequent misclassification of shallow bare zones as shallow water, which is observed in the SVM model. This performance may be explained by the depth of the RF model with the large number of splits allowing it to classify well. This along with the smaller number of trees in the forest, this may hurt the RF model's ability to generalize to other data.

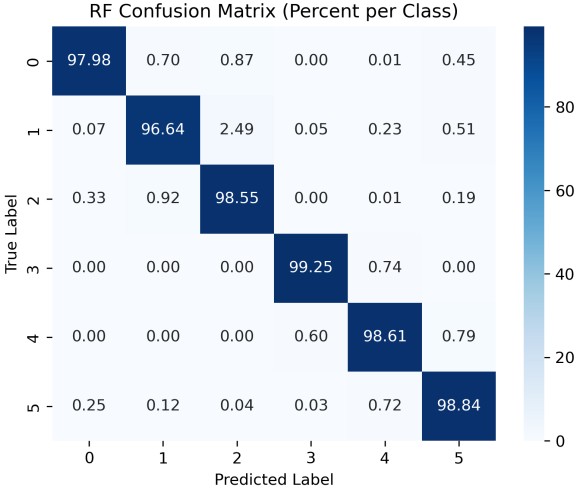

**Figure 6.** RF confusion matrix on the independent test set. Excellent performance is observed across all classes.

Some aspects in which the SVM model has clear advantages over the RF in regards to prediction time, and file size which are important factors to consider of live image classification. The training of the SVM model took 20 minutes to train and was able to classify the test set in as fast as 0.51 seconds. Besides this, its minimal file size of 1.28 Kilobytes allows it to be utilized by microcontroller

devices such as the Arduino line of microcontrollers which have minimal storage space. This opens up the possibility for live image prediction. In reality, a real-time imaging system would take in images smaller than those used in the test set thus having a quicker effective classification time. In comparison, the RF model trains slower needing nearly an hour to train and falls behind in other metrics with a prediction time 10 times longer than the SVM and a file size three and a half orders of magnitude larger. Though the prediction time of the RF model may still be considered usable in some cases, the model is best suited to accurate post-processing in which file size and prediction time are not much of a concern. It would be likely of the three microcontrollers mentioned in this work's introduction that the SVM would be usable on all three whereas the RF model would only likely find success when implemented on the Raspberry Pi. Listed in Table 3. is a summary of the major differences between the two models.

**Table 3.** Summary of differences between optimized SVM and RF models. RF is suited for post processing, SVM shows potential for live classification tasks.

| | SVM | RF |
|---|---|---|
| Accuracy | 0.85 | 0.98 |
| Precision (Macro) | 0.86 | 0.92 |
| Recall (Macro) | 0.85 | 0.91 |
| F1-Score (Macro) | 0.85 | 0.91 |
| Training Time (s) | 1259 | 3244 |
| Prediction Time (s) | 0.51 | 4.957 |
| File Size (KB) | 1.28 | 1890 |

# 5 Conclusion

This study demonstrates the effectiveness of ML methodologies in classifying coastal terrain using multispectral images captured by a UAV in tropical coastal regions. By implementing K-means clustering for initial terrain identification and training SVM and RF models, the research identified RF as the superior model for this application, outperforming SVM across most metrics. Despite this, the optimized SVM model showed promise for live classification due to its smaller size and quicker prediction time. The successful classification of images from areas in the test set underscores the further applicability of ML techniques in remote sensing. These findings reinforce RF models in providing robust ML frameworks for accurate classification. At the same time, SVMs are seen to have potential in terrain classification in resource-constrained environments. Future research could explore the application of these methods to other geographic regions and further optimize the models for broader

use in remote sensing.

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
