# OpenReview forum: "Machine Learning-Based Coastal Terrain Classification in Tropical Regions Using Multispectral UAV Imaging: A Comparative Study of Random Forest and SVM Models"
_NLDL.org/2025/Conference — Submitted to NLDL 2025_

### Official Review · Reviewer_cYsE · 2024-09-16
**Dataset represents trivial task, not deep learning**

**Confidence:** 4

**Summary:**

The paper first introduces a new dataset consisting of multi-spectral UAV images, captured in a coastal area of the Philippines. The purpose of this dataset is to investigate different methods for pixel-wise classification of these images into one of several ecological classes. The authors then train and compare SVMs and random forest models on this dataset.

**Strengths:**

1. The collection of a new dataset in an ecologically relevant domain is the biggest strength of this paper. Will the dataset be released?
2. The motivation for using UAVs is very clear.
3. The description of the data collection and anything related to the area of remote sensing was clear and gives the impression that the authors are knowledgeable in this area.
4. The methods used are well-known, using cross-validation is a standard good practice to ensure that experimental results are reliable. See "weaknesses" for a possible caveat on this.

**Weaknesses:**

1. The authors use a cross-validation approach combined with grid search to find the best hyperparameter settings, describing the sets used as training and validation set. Later, they talk about a test set. I'm not sure if that was kept separate (if so, it does not seem to be mentioned), or if they use the validation set from the cross-validation and just call it test set here. In the latter case, the hyperparameters would have been chosen based on the test set during the grid search, which would go against best practices.

2. I'm not completely sure if I understand correctly how the labels were set. If the labels are based only on k-means clustering (choosing the number of clusters via silhouette score), k-means on the training set should be used as a baseline method. Comparing to k vectors is very performant, and should perform very well, if the original labels are determined via k-means. However, this would also mean that trying to predict those labels would not be very challenging, which explains the good scores of the RF model. Manual inspection and postprocessing of the labels, on the basis of k-means, could lead to better results and a more interesting dataset. In its current form, the problem seems trivially solvable.

3. The authors mention that they want to "identify suitable ML algorithms for analyzing multispectral images on limited hardware." I think that's a good goal, I could imagine that there is some kind of onboard processing on the drone while it's capturing images. If that's the case, the requirements that this goal of onboard processing imposes on the models should specified more clearly than just "limited hardware". E.g.: How big can a model be in memory? What is this limited hardware? If this is not specified, it is not possible to judge the results adequately with respect to this goal.
The prediction time in Table 3 would also be very informative if it was somehow related to the onboard prediction time. Given that the random forest takes 15 seconds for a prediction, I assume that the forest or the batch size are rather large. A large batch size is not something to be expected onboard the drone. A large forest, or deep trees, could be trimmed to trade off prediction time for performance. But this all relies on knowing more specific conditions.

4. The authors mention how big the region is that they collected the data in, and the resolution of the images, but not the final size of the dataset, e.g. in number of pixels.

5. Figures 1 to 5 are never referred to in the text.

6. In Figures 4 and 5, I don't know what the different classes are without switching back and forth to the place where they're mentioned. Replacing the integer labels with the class names would make this much more accessible to the reader.

7. Given that the submission concerns a "deep learning conference", I think the paper should use deep learning methods, which it currently does not.

**Final Rebuttal Confidence:**

4

**Final Rebuttal Justification:**

I commend the work that the authors have put in to respond to the reviewers' comments. The paper has clearly improved after the initial reviews: It is now much clearer how labels were created, and on which data the models were evaluated (Fig. 2). Including information about possible hardware and its limitations also makes the on-board-processing argument stronger. Mentioning the number of pixels included was also important.

My remaining concerns are of the kind that is unfortunately hard to address during a rebuttal phase.

1. The paper does not use deep learning methods, but is submitted to a deep learning conference. In a cursory search, I have not found any papers at past versions of this conference which did not include elements of deep learning, though I may have missed some. While the authors rightfully mention that topics like supervised learning are suggested under "General Machine Learning", that point is mentioned right under the headline of "We invite submissions presenting new and original research on all aspects of Deep Learning." I understand this to mean that the general machine learning topics of supervised or active learning are of interest, insofar as they pertain to deep learning methods, as well as topics like architectures, which are topics specific to deep learning.

2. Since this is an application paper, the review guidelines ask that the problems be non-trivial, and the solutions novel, effective and/or practically relevant. Given that a random forest almost perfectly solves the presented problem, the problem appears to be trivial. The solution is also not novel, but a well-known standard method. I am not a domain expert in this domain of ecology, so I can not judge whether the result is of practical relevance in this field. I think this paper would be better suited for a venue that focuses on remote sensing or the specific ecological domain, where the respective domain experts can make this decision of relevance.

**Justification:**

This paper is clearly an application paper, for which the review guidelines ask that the problems be non-trivial, and the solutions novel, effective and/or practically relevant. Given that a random forest almost perfectly solves the presented problem, the problem appears to be trivial. The solution is not novel, but a well-known standard method. The solution may be effective and practically relevant, but I do not count this towards the paper's strengths, because it is an existing standard method in the field.

I am unfamiliar with the literature on this specific problem but it might be that being able to perform the presented classification this well is a step forward in terms of domain-specific knowledge. In that case, I believe that remote sensing venues would be a much more fitting place for this paper, since it does not provide any value in the area of deep learning.

Using labels based on k-means clustering for supervised learning does not seem like a reasonable approach to me, since the resulting labels are very easy to predict, as demonstrated by the experimental section. If the labels do not provide a meaningful challenge, the experimental results also do not provide any advancement in knowledge, apart from the task being trivial.

In summary, the paper presents a dataset which can be almost perfectly solved by a standard ML method, does not provide any innovation in terms of methodology, and does not belong into the area of deep learning, which is why I a recommend a rejection.

The value that the paper provides is the new dataset. To make this a stronger paper, I would recommend publicizing the dataset and finding tasks that can be solved with it, that are relevant to the application domain and non-trivial. Focusing more on the aspect of onboard processing, with concrete restrictions which the methods have to fulfill, could also be an interesting approach.

---

> ### Author Rebuttal · Authors · 2024-10-25
>
> We appreciate your valuable feedback and suggestions. We have carefully considered all the comments and made revisions to the manuscript accordingly. Below, we provide detailed responses to each comment:
>
> ---
> **Comment:**
> *"The collection of a new dataset in an ecologically relevant domain is the biggest strength of this paper. Will the dataset be released?"*
>
> **Response**
> There are no intentions by the authors to release the dataset as of the writing of this response. However, the dataset may be released at a later date.
>
> **Comment:**
> *"The authors use a cross-validation approach combined with grid search to find the best hyperparameter settings, describing the sets used as training and validation sets. Later, they mention a test set. It’s unclear if that was kept separate; if so, it is not explicitly stated. If they use the validation set from the cross-validation and refer to it as the test set, this would mean that hyperparameters were chosen based on the test set during grid search, which goes against best practices."*
>
> **Response:**
> The original manuscript referred to the test set as the validation set from the most successful model in the k-fold cross-validation. The revised manuscript clarifies the approach taken to validate the models, including the implementation of a block k-fold cross-validation scheme alongside a hold-out test set.
>
> **Comment:**
> *"I’m unsure if I fully understand how the labels were established. If they are based solely on k-means clustering (with the number of clusters determined via silhouette score), then k-means on the training set should serve as a baseline method. Comparing to k vectors is very efficient and should yield strong performance if the original labels stem from k-means. However, this would suggest that predicting those labels may not be very challenging, which could explain the strong performance of the RF model. Manual inspection and post-processing of the labels based on k-means could yield better results and a more interesting dataset. As it stands, the problem appears trivially solvable."*
>
> **Response:**
> The labels are not solely derived from k-means clustering; they undergo a process of manual inspection and refinement. Nevertheless, the labels remain closely aligned with those generated by k-means, as this methodology was chosen to expedite the labeling process.
>
> **Comment:**
> *"The authors state their aim is to 'identify suitable ML algorithms for analyzing multispectral images on limited hardware.' While this is a commendable goal, I wonder if there is potential for onboard processing on the drone while capturing images. If so, the manuscript should clearly specify the requirements imposed by this onboard processing beyond just 'limited hardware.' For example, what is the memory limit for a model? What exactly does 'limited hardware' refer to? Without this information, it is difficult to adequately evaluate the results in relation to this goal. Additionally, relating the prediction times in Table 3 to the onboard prediction times would be informative. Given that the random forest takes 15 seconds for a prediction, I assume the forest or batch size is relatively large. A large batch size is unlikely to be feasible onboard the drone. A large forest or deep trees might need to be trimmed to balance prediction time with performance, but this depends on understanding the specific conditions."*
>
> **Response:**
> In response to this comment, we have included further details about potential onboard processing capabilities in the Introduction and Discussions sections. Additionally, we conducted further analysis on the trees, focusing on their depth and width. This led to the selection of a different RF model that sacrifices a small degree of accuracy and training time in exchange for faster prediction times. The revised manuscript also discusses suitable microprocessors for these models.
>
> **Comment:**
> *"The authors mention the size of the region from which the data was collected and the resolution of the images but do not specify the final size of the dataset in terms of the number of pixels."*
>
> **Response:**
> The total number of pixels in the dataset is approximately 50,000,000. This detail has been added to Chapter 2.2 of the revised manuscript.
>
> **Comment:**
> *"Figures 1 to 5 are not referenced in the text."*
>
> **Response:**
> All figures have now been referenced in the text as per this suggestion.
>
> **Comment:**
> *"In Figures 4 and 5, the different classes are not clear without referencing back to where they are mentioned. Replacing integer labels with class names would improve accessibility for readers."*
>
> **Response:**
> We have replaced integer labels with class names in the figures where it does not lead to excessive clutter.
>
> **Comment:**
> *"Given that the submission pertains to a 'deep learning conference,' I believe the paper should employ deep learning methods, which it currently does not."*
>
> **Response:**
> While it is correct that the submission does not utilize deep learning methods, the list of recommended topics includes General Machine Learning, defined as "General Machine Learning (active learning, clustering, online learning, ranking, reinforcement learning, supervised, semi- and self-supervised learning, time series analysis, etc.)."
>
> ---
>
> We hope these revisions effectively address your concerns. Please let us know if you need any further clarification. Thank you once again for your insightful comments and suggestions.

---

### Official Review · Reviewer_v4h7 · 2024-09-22
**Comparison of RF and SVM on k-means clustered Vegetation Indices**

**Confidence:** 3

**Summary:**

In brief the paper describes a pipeline:
- UAV images are processed using onboard capabilities, to genereate several different VI (vegetation indices)
- Labels are generated using k-means, with a k between 2 and 8. (further in the paper it seems to resolve as 6 classes)
- Random Forest and Support Vector Machines are fitted using a k-fold validation split.
- Comparison between the two seem to indicate a stronger fit using Random Forest, albeit at a cost of computation time.

The paper purports to be a demonstration into the effectiveness of ML for classifying coastal areas, but there remains some unclear aspects to this research.

**Strengths:**

- The paper is clear and logically structured, describing the approach in an easy to follow manner, with clear figures and tables.
- The experiments are clear, but the description of the data and testing is a bit lacking.
- Explanations of methods used are clear, and the methods for classification are well founded within the field. e.g. RF being a staple for performing pixel-wise classification of nature-types.
- The paper demonstrates an interesting way to tune pixel-wise classifiers that correspond well with clusters identified using k-means, and may therefore be useful for other studies that are planning longer term UAV imaging over a constrained/well-known area.

The overall pipeline of unsupervised clustering to support training SVM or RF classifiers seems reasonable, and worth a qualitative evaluation, especially for use with onboard capabilities of an UAV.

**Weaknesses:**

**Missing/Bigger stuff**:

- The paper does not describe how the authors have performed a train/test split, with a hold out test set that they report on. There is mention of a test-image, but this should be made explicit and clear in the paper, otherwise it is tempting to assume that the reported numbers reflect the validation score from the k-fold-validation.
- It would be useful to mention or compare with other similarly used methods, e.g. kNN, Bayes, or xg-boost, but it is pointed out in the paper that the authors seemed to be operating on limited computation budget.
- When the labels are generated using a clustering algorithm like k-means, unless these data are manually verified and adjusted, it seems likely that the comparison performed is one of "How similar is RF/SVM to k-means?". This could probably be clarified better in the paper, along with a justification as to how the labels are qualitatively evaluated. It is perhaps not suprising that RF would be a good fit for separating members of clusters.
- Experiments with smaller forests, since time is mentioned as a constraint.

**Errata/smaller stuff**:
- F1 score should have a capital F (line115)
- LiDAR should have a lower case "i" (line 36)
- line 54-56, "its performance relies ... on" should probably be "their performance rely ... on" (refering to the class of SVMs earlier as they)
 - K-fold cross-validation is also a tunable parameter.
- on line 256; "plurality of pixels" seems better stated as "majority of pixels"
- only a partial list of the bands in the UAV seems to be included.
- missing the max depth of random forest and the width / number of trees. This might be desirable to report w.r.t. time being a constraint.

**Final Rebuttal Confidence:**

4

**Final Rebuttal Justification:**

This paper seems focused on the application of machine learning, as it pertains to a potential on-board capability of UAVs.
It follows a reasonable, standard methodology for setting up the capabilities described, and it does not seem to introduce any novel approach to the pipelines it outlines.

The authors describe well how they've performed the work, demonstrating that one can use RF to perform segmentation of image-data retrieved from an UAV. They also demonstrate that RF has some advantages over SVMs when it comes to performance in this case, and conclude that machine learning is effective at the proposed task.

Beyond its value as a clear report of an implementation of ML as a possible future on-board pixel-based image segmenter, the paper provides no novel perspective or implementation details. The details describing the constraints of the UAVs, and what the authors suggest would be suitable hardware to support their ML-model, any mention of fitting their approach to these constraints is also lacking.
i.e. it is still unclear whether the task they suggest; The use of ML as an on-board / live image segmentation tool, is feasible within their constraints. (It likely is, yet the authors never return to their stated intentions for performing this work)

It should be mentioned that the authors seem to have taken the feedback given to heart, and improved their paper substantively from the initial submission.
As the paper stands now it is clear what, and how, the work reported has been performed.
There remains some minor grammatical mistakes, but these do not presently affect the final assessment.

**Justification:**

- Lacking a clear description of Test-set/-data makes it difficult to trust the reported numbers. As the authors have pointed out, RF is liable to overfit, and without being able to verify that the Test-data is clearly separated from the training data, the reported numbers may be tainted.
- Would like some more justification on what the result of fitting RF/SVM to k-means clusters is. It seems here that what is outlined in the paper is a way of tuning on-board capable algorithms to perform what would otherwise be done post-hoc using clustering algorithms. This may have futher utility, but the degree to which it is a novel contribution is unclear. Tuning on-board capable classifiers is an active field within remote-sensing, but the label data will usually include human experts. An unsupervised approach may provide more usable data, but the evaluation of results should probably then include domain experts who validate the k-means generated clusters before these are trained on.

---

> ### Author Rebuttal · Authors · 2024-10-25
>
> We are grateful for your valuable feedback and suggestions. We have carefully considered all the comments and made revisions to the manuscript accordingly. Below, we provide detailed responses to each comment:
>
> ---
>
> ### **Missing/Bigger Issues:**
>
> **Comment:**
> _"The manuscript does not clearly describe how the authors performed the train/test split, including the hold-out test set they report on. While there is a mention of a test image, this should be explicitly stated to avoid assumptions that the reported numbers reflect the validation score from k-fold cross-validation."_
>
> **Response:**
> We have revised this section for clarity. The methodology has been updated to explicitly detail how the hold-out test set is separated from the train-validation section.
>
> **Comment:**
> _"It would be useful to mention or compare with other methods such as kNN, Bayes, or XGBoost. However, the authors note that they are operating under a limited computational budget."_
>
> **Response:**
> While this suggestion could provide valuable insights, the limited computational resources are a significant factor in the decision not to include other models.
>
> **Comment:**
> _"When generating labels using a clustering algorithm like k-means, unless these data are manually verified and adjusted, it seems likely that the comparison is essentially assessing 'How similar is RF/SVM to k-means?' This should be clarified, along with a justification for how the labels are qualitatively evaluated."_
>
> **Response:**
> After clustering, each cluster is manually inspected and adjusted by the authors to accurately represent the terrain types. The revisions aim to clarify this process through enhanced text and additional figures.
>
> **Comment:**
> _"Experiments with smaller forests are suggested, especially since time is noted as a constraint."_
>
> **Response:**
> Chapter 4.1 in the Discussions section has been added to elaborate on the number and depth of trees used in the RF model. It was found that tree depth significantly influences model performance.
>
> ---
>
> ### **Errata/Minor Issues:**
>
> **Comment:**
> _"The F1 score should be capitalized (line 115)."_
>
> **Response:**
> This has been corrected as suggested.
>
> **Comment:**
> _"'LiDAR' should have a lowercase 'i' (line 36)."_
>
> **Response:**
> This has been corrected as suggested.
>
> **Comment:**
> _"In lines 54-56, 'its performance relies ... on' should probably be changed to 'their performance relies ... on' (referring to the class of SVMs as 'they')."_
>
> **Response:**
> This has been revised as suggested.
>
> **Comment:**
> _"K-fold cross-validation is also a tunable parameter."_
>
> **Response:**
> While this is correct, the authors chose not to tune this parameter due to time constraints and limited computational power.
>
> **Comment:**
> _"In line 256, 'plurality of pixels' seems better stated as 'majority of pixels.'"_
>
> **Response:**
> This has been revised as suggested.
>
> **Comment:**
> _"Only a partial list of the bands in the UAV appears to be included."_
>
> **Response:**
> The manuscript includes a complete list of bands in Chapter 2.1.
>
> **Comment:**
> _"The manuscript is missing the maximum depth of the random forest and the number/width of trees, which may be relevant given time constraints."_
>
> **Response:**
> While the revised manuscript does not provide specific details on the influence of random forest hyperparameters, it does mention the general effects of these parameters on model performance and training time.
>
> ---
>
> We hope these revisions satisfactorily address your concerns. Please let us know if you require any further clarification. Thank you again for your insightful comments and suggestions.

---

### Official Review · Reviewer_Wjcx · 2024-10-09
**The motivation behind the paper is overall clearly presented and contextualized. The goal of the work is also carefully and clearly described. The architecture and the metrics selected are reliable.  Big limits on degree of novelty, data collection and validation.**

**Confidence:** 4

**Summary:**

The motivation behind the paper is clearly presented and contextualized. The goal of the work is also carefully and clearly described. The paper identifies a bottleneck in UAV and investigates solutions to implement real/near-real-time image classification. The architectures and the metrics selected are very reasonable and reliable.

The introduction and the conclusions are well-written, with just one thing not explained. While the issue is firstly clearly explained, it is not clear why it is important to classify real-time UAV images and how difficult the challenge is.

Nevertheless, the paper has several weaknesses. Firstly, the lack of novelty. There is plenty of literature applying ML to UAV imaging for the classification of the natural environment, including forestry, wetlands, cultivated areas, or coastal terrains. Several of them also aim at a semi-real-time classification. Secondly, the training data have been collected only within a very small area. More data should have been collected from different locations and in different conditions, in order to improve the amount and quality of the training dataset. Thirdly, the methodology followed to create the labels isn't clear. Finally, an extensive validation is needed. Indeed, the biggest weakness of the work is that the models have been validated in the same area where they have been trained. Validation in a new area is needed to consider the results reliable and publishable.

**Strengths:**

As mentioned in the summary, the motivation behind the paper is clearly presented and contextualized overall, with a clear introduction to the topic. The goals of the work are also clearly described.

Other strengths of the work:
- The need for AI is well-motivated.
- The model and architecture selected are standard but reliable ones. SMV and RF are two of the most popular and reliable choices when dealing with multispectral imagery. The specifications of the architecture of the model and its choice are well-motivated.
- The metrics selected are meaningful and reliable. A variety of metrics have been selected and the results are clearly presented using several tables.
- The conclusion is clearly formulated.

**Weaknesses:**

The review identified several, substantial weaknesses:

-  Lack of novelty. There is plenty of literature applying ML to UAV imaging for the classification of the natural environment, including forestry, wetlands, cultivated areas, or coastal terrains. A lot of it is also aiming on a near-real-time classification of the images.
- Data collection. The data collected are very limited, all within a single campaign in a small area. More data should have been collected from different locations and in different conditions, in order to improve the amount and quality of the dataset. Info on the conditions during which the data are absent.
- Labels: the methodology followed to create the labels isn't clear.
- Validation: this is definitely the biggest weakness of the work. Indeed, the models have been validated in the same small area where they have been trained. Taking 80% of the data within an 8-hectare area for training and 20% for validation and going through training-validation cycles isn't the same as validating on a different area, or randomly selecting an area on which the model hasn't been trained. Perform a real independent validation on a new area, which is considered necessary for considering the paper for publication.

Regarding the structure of the publication:
- The results and their discussion should be more clearly separated.
- The figures need a step forward in terms of explanation and quality.

More in detail:

Abstract
- The abstract doesn't mention one of the main findings of the work: that SVM is better suitable for live clarification. Add it.

Chapter 1
- While the issue is firstly clearly explained, it is not clear why it is important to classify real-time UAV images and how difficult the challenge is. Please elaborate, making it clear.

Chapter 2.1
- Why data have been collected only in one (and such a small) area? When (season, light conditions)? In a single campaign or in several? For reliable training, a few more data campaigns should have been carried out: one of them fully dedicated to validation.

Chapter 2.2
- Line 98-100: Add a figure of one of those orthomosaics.
- Are there any artifacts present in the orthomosaics? This sometimes could be an issue.

Chapter 2.3
- The chapter isn't well written. Try to rephrase, focusing on elaborating on how the labels were classified.
- Line 129: Add reference on silhouette score.

Chapter 2.4
- Remove "of ML" from the title.

Chapter 3

It would be better to have the presentation of the results and their discussion in two separate chapters.

Chapter 3.1
- Reading the chapter, it seems that the classes described are the outcome of the models. My understanding is that those are the labels used to train the models. What is correct? If the latter, then those aren't results and shouldn't been included in a chapter presenting and discussing the results. If they are results, then it needs to be explained much better.

Chapter 3.2
- The chapter is well written, but the issue is how the validation has been carried on: see what is written above.
- Line 218-221: this sentence is useless. Please remove.
- Lines 262-265: Any idea why the RF model largely prevents the frequent misclassification of shallow bare zones as shallow water, which is observed in the SVM model? Please add a sentence elaborating on the possible cause of this misclassification.
- Line 274-277: It is true that the prediction time would be shorter, but what about the resources available for real-time (live) classification? Do they have an impact?

Chapter 3.3
- This section presents results, so it goes before the discussion presented in Chapter 3.2.
- Line 293-295: Rephrase because it is not clear, and point to the figure.

Figures and Tables: All figures' captions need to be extended, containing a clear description of what the figure is. When colours are present, their meaning needs to be clearly explained. Each caption should also contain 1 sentence highlighting the main "take-home" message that the author wants the reader to remember.

- Figure 3: Improve the resolution of the images.
- Table 1: Elaborate pointing out highlights and explaining why some numbers are in red.
- Table 2: See comments for Table 1.
- Figure 6: Elaborate, the caption doesn't allow to understand what the figure is showing.
- Figure 7: What is written on the axis and the percentages are unreadable. Increase the size of the font.
- Table 3: Why green? See comments for Table 1.

**Justification:**

I believe the paper isn't ready to be published. The first reason is that it lacks novelty. Indeed, there is plenty of literature already available applying ML to UAV imaging for the classification of the natural environment, including forestry, wetlands, cultivated areas, or coastal terrains. Even in the lack of novelty, it can be considered for publication because it addresses a specific bottleneck in the application of ML to UAV, but only after more work has been performed in order to back up the results and assess the weaknesses identified during the review. The weaknesses are the following:

- Data collection only took place within a small area. This strongly limits the applicability of the model to other areas/terrains. More data need to be collected from different locations and in different conditions, in order to improve the amount and quality of the dataset. The model should be retrained using this updated dataset.

- Validation: this is definitely the biggest weakness of the work described. Indeed, the models have been validated in the same small area where they have been trained. Taking 80% of the data within an 8-hectare area for training and 20% for validation and going through training-validation cycles isn't the same as validating on a different area, or randomly selecting an area on which the model hasn't been trained. Performing a validation in an area that the model hasn't been trained on is necessary for considering the paper ready for publication.

---

> ### Author Rebuttal · Authors · 2024-10-25
>
> Dear Reviewer,
>
> We are grateful for your valuable feedback and suggestions. We have carefully considered all the comments and made revisions to the manuscript accordingly. Below, we provide detailed responses to each comment:
>
> ---
>
> ### **Weaknesses:**
>
> **Comment:**
> *"Lack of novelty. There is plenty of literature applying ML to UAV imaging for the classification of the natural environment, including forestry, wetlands, cultivated areas, or coastal terrains. Much of it is also aimed at near-real-time classification of the images."*
>
> **Response:**
> We appreciate the reviewer's observation regarding the existing literature on this topic. However, we wish to highlight the novel application of these methodologies in the context of the Philippines and Southeast Asia. Furthermore, our approach utilizes k-means clustering to create training labels derived from vegetation indices, which streamlines the labeling process and contributes a unique perspective to the field. It is also important to note that while much of the current research on near-real-time classification predominantly examines RGB images, our study specifically addresses the classification of hyperspectral images.
>
> ---
>
> **Comment:**
> *"Data collection. The data collected are very limited, all within a single campaign in a small area. More data should have been collected from different locations and under varying conditions to improve the dataset's amount and quality. Information on the conditions during data collection is missing."*
>
> **Response:**
> We recognize this limitation in the dataset used in the study. Additional details regarding the conditions of data acquisition have been added to the manuscript.
>
> ---
>
> **Comment:**
> *"Labels: the methodology followed to create the labels isn't clear."*
>
> **Response:**
> We have revised the manuscript to clarify the methodology used to create the labels. Additional figures have been included to further enhance the explanation.
>
> ---
>
> **Comment:**
> *"Validation: this is the biggest weakness of the work. The models were validated in the same small area where they were trained. Simply splitting the data within an 8-hectare area for training and validation does not provide robust results. A true independent validation on a new area is necessary."*
>
> **Response:**
> We agree that the validation approach could be strengthened. To address this, we revised the validation methodology by implementing a block k-fold cross-validation, along with a hold-out test set, to better assess the model's performance. However, we also acknowledge that additional data from other locations would further improve the validation process.
>
> ---
>
> ### **Abstract:**
>
> **Comment:**
> *"The abstract doesn't mention one of the main findings of the work: that SVM is better suited for live classification. Add it."*
>
> **Response:**
> We have revised the abstract to include the SVM model's potential suitability for live classification tasks.
>
> ---
>
> ### **Chapter 1:**
>
> **Comment:**
> *"While the issue is clearly explained, it is not clear why real-time UAV image classification is important or how challenging it is. Please elaborate."*
>
> **Response:**
> We have added a paragraph in Chapter 1 elaborating on the significance and challenges of real-time UAV image classification.
>
> ---
>
> ### **Chapter 2.1:**
>
> **Comment:**
> *"Why was data collected only in one small area? When (season, light conditions)? Was it collected during a single campaign or several? More data campaigns are needed, with one dedicated to validation."*
>
> **Response:**
> We acknowledge this limitation. Details regarding the data acquisition process, including the time of year, light conditions, and campaign information, have been added to Chapter 2.1. Additionally, we have revised Chapter 2.5 to discuss the updated validation techniques.
>
> ---
>
> ### **Chapter 2.2:**
>
> **Comment:**
> *"Line 98-100: Add a figure of one of the orthomosaics."*
>
> **Response:**
> We have added a figure of the orthomosaic, now included as Figure 1.
>
> **Comment:**
> *"Are there any artifacts present in the orthomosaics? This can sometimes be an issue."*
>
> **Response:**
> We applied a median filter to assess potential artifacts in the orthomosaics. No significant artifacts were found, though slight transitions at the image boundaries were detectable with high threshold values. This has been explained in the manuscript.
>
> ---
>
> ### **Chapter 2.3:**
>
> **Comment:**
> *"This chapter isn't well written. Please rephrase it and focus on explaining how the labels were classified."*
>
> **Response:**
> We have rewritten this section for clarity, with a more detailed explanation of the label classification process. A new figure (Figure 2) has also been added to illustrate the label generation methodology.
>
> **Comment:**
> *"Line 129: Add a reference on the silhouette score."*
>
> **Response:**
> We have added the appropriate reference for the silhouette score.
>
> ---
>
> ### **Chapter 2.4:**
>
> **Comment:**
> *"Remove 'of ML' from the title."*
>
> **Response:**
> We have modified the title as requested.
>
> ---
>
> ### **Chapter 3:**
>
> **Comment:**
> *"It would be better to present the results and discussion in separate chapters."*
>
> **Response:**
> We have made this change. The results are now presented in Chapter 3, and the discussion has been moved to Chapter 4.
>
> ---
>
> ### **Chapter 3.1:**
>
> **Comment:**
> *"It seems that the described classes are the outcome of the models. If these are training labels, they should not be in the results section. Please clarify."*
>
> **Response:**
> You are correct. These are the labels generated by k-means clustering to train the RF and SVM models. We have moved this information to Chapter 2.
>
> ---
>
> ### **Chapter 3.2:**
>
> **Comment:**
> *"Line 218-221: This sentence is unnecessary. Please remove."*
>
> **Response:**
> We have removed this sentence as requested.
>
> **Comment:**
> *"Lines 262-265: Why does the RF model largely prevent the misclassification of shallow bare zones as shallow water, which is observed in the SVM model? Please elaborate."*
>
> **Response:**
> We believe this is due to the depth and structure of the trees in the random forest model. This explanation has been added to the manuscript.
>
> **Comment:**
> *"Line 274-277: You mention prediction time, but what about the impact of resource constraints on real-time classification? Please elaborate."*
>
> **Response:**
> We have expanded the discussion on resource constraints, including RAM, storage capacity, and power consumption, which are critical factors for live classification on UAVs.
>
> ---
>
> ### **Chapter 3.3:**
>
> **Comment:**
> *"This section presents results, so it should be before the discussion in Chapter 3.2."*
>
> **Response:**
> We have reorganized the manuscript, moving this section as suggested.
>
> **Comment:**
> *"Line 293-295: Rephrase for clarity, and point to the figure."*
>
> **Response:**
> We have rephrased this section and explicitly referenced the corresponding figure.
>
> ---
>
> ### **Figures and Tables:**
>
> **Comment:**
> *"All figures' captions need to be extended with clearer descriptions and explanations for colors. Include a key message in each caption."*
>
> **Response:**
> We have extended the captions for all figures and tables, clarifying the content and explaining the use of colors. We have also added a key takeaway message for each.
>
> **Comment:**
> *"Figure 3: Improve the resolution of the images."*
>
> **Response:**
> We have replaced the images with higher-resolution versions.
>
> **Comment:**
> *"The caption doesn't allow understanding what the figure is showing. Please elaborate."*
>
> **Response:**
> The caption has been elaborated to clarify the content of the figure.
>
> **Comment:**
> *"Figure 7: The axis labels and percentages are unreadable. Increase the font size."*
>
> **Response:**
> This figure has been removed from the revised manuscript.
>
> ---
>
> We hope these revisions address your concerns. We are happy to provide further clarification if necessary. Thank you again for your insightful comments and suggestions.

---

### Official Review · Reviewer_mNAm · 2024-10-09
**Comparison of SVM and Random Forest for Coastal Terrain Classification: Lacking Novelty and Comparison with State-of-the-Art Techniques**

**Confidence:** 5

**Summary:**

This paper presents a comparison between Support Vector Machines (SVM) and Random Forest (RF) for coastal terrain classification (segmentation). The authors used a dataset with 2 cm resolution, collected using UAVs, and applied SVM and RF to orthomosaics that combine multiple spectral bands.

**Strengths:**

The paper is well-written and easy to follow.
The authors provide a clear comparison between SVM and RF for pixel-level classification.
They gave a detailed description of the methodology and data preparation.

**Weaknesses:**

The authors merely applied SVM and RF to their dataset without introducing new techniques.
There is no comparison with state-of-the-art semantic segmentation methods.
The paper lacks a clear contribution beyond the application of existing machine learning algorithms.
The analysis of the results is superficial, focusing on describing the results without providing any justification for them.

**Justification:**

The paper is well-written and presents a comparison between SVM and RF for coastal terrain classification. However, it lacks a significant contribution beyond the application of standard machine learning algorithms to the custom dataset. Furthermore, the authors did not compare their methods with state-of-the-art semantic segmentation techniques.

---

> ### Author Rebuttal · Authors · 2024-10-25
>
> We appreciate your valuable feedback and suggestions. We have carefully considered all the comments and made revisions to the manuscript accordingly. Below, we provide detailed responses to each comment:
>
> ---
>
> **Comment:**
> *"The authors merely applied SVM and RF to their dataset without introducing new techniques. There is no comparison with state-of-the-art semantic segmentation methods."*
>
> **Response:**
> The authors did not consider other semantic segmentation methods due to limitations in computational power. However, we acknowledge that such comparisons would be beneficial for contextualizing our work within the existing literature.
>
> **Comment:**
> *"The analysis of the results is superficial, focusing on describing the results without providing any justification for them."*
>
> **Response:**
> The revised manuscript now offers a more in-depth discussion of how the models were optimized and provides possible explanations for the observed model performance in the newly added Discussions chapter.
>
> ---
>
> We hope these revisions satisfactorily address your concerns. Please let us know if you require any further clarification. Thank you once again for your insightful comments and suggestions.

---

### Meta-Review · Area_Chair_aSEx · 2024-11-01

**Recommendation:** Reject
**Confidence:** 3

**Metareview:**

The paper discusses an interesting topic of using machine learning to identify coastal terrain acquired by UAV multispectral sensors. Reviewers rise a number of questions, with a general negative trend regarding the methodology novelty (application of existing and old methods to a new dataset), the validation and discussion of results, and the lack of more recent, deep-learning based, baselines. Despite the additional work and improvments proposed by Authors in the revised version, acknowledged by reviewers, it remains that the NLDL conference might not be the best choice as a publication venue for this paper, even as an application paper.

**Suggested Changes To The Recommendation:**

3: I agree that the recommendation could be moved up

---

### Decision · Program_Chairs · 2024-11-06

Reject